# Lean4trace: Data augmentation for neural theorem proving in Lean

**Vasilii Nesterov** [1]  **Yermek Kapushev** [2]  **Mikhail Burtsev** [3]

## Abstract

Integrating large language models as proof assistants with theorem provers has shown great promise. However, one of the major challenges in this field is the scarcity of training data. To address this, we release a new open-source tool, *Lean4trace*, for training data extraction from Lean 4 sources. Unlike previous approaches, *Lean4trace* is deeply integrated into the Lean elaborator, allowing us to modify proofs on-the-fly. Leveraging this feature, we propose two methods of data augmentation in Lean: (1) decomposing composite proof steps into multiple simpler steps; (2) testing existing proof automation tactics at each proof state and collecting the successful ones. Models trained on this augmented data are capable of proving 58.0% of theorems from a hold-out subset of Mathlib and 35.6% of the test subset of the MiniF2F benchmark.

## 1. Introduction

One of the advantages of mathematics over other sciences is that the correctness of its results can, in principle, be verified mechanically. This is particularly desirable because the standard peer review process inevitably sometimes results in invalid proofs. However, in practice, formalizing mathematics is a labor-intensive and time-consuming task. The standard approach involves using interactive theorem proving (ITP) systems, among which it is worth mentioning Lean (de Moura et al., 2015), Isabelle (Nipkow et al., 2002), Coq (Barras et al., 1997), and Metamath (Megill & Wheeler, 2019). The process of theorem proving in such a system is similar to programming in an IDE: a user interacts with the system via commands in a formal language, and the system provides feedback on whether the proof is successful.

In recent years, significant efforts have been made to simplify the formalization process. The most developed libraries of formalized mathematics now contain more than 100,000 theorems. One example is Mathlib (mathlib Community, 2020), a user-maintained mathematical library for the Lean theorem prover, which covers a wide range of mathematical fields.

From another perspective, formal theorem proving, as well as reasoning in general, remains a significant challenge for AI systems. Recently, several approaches to this task have been proposed, all of which are based on transformer (Vaswani et al., 2023) language models. Modern natural language models are typically trained on large corpora of data, while the training data extractable from proofs is relatively scarce. In our paper, we address this challenge by proposing two methods of data augmentation specific to the task of formal theorem proving.

In our study, we focus on the Lean[1] theorem prover as an ITP system and Mathlib as the main source of training data. Theorems in Lean are typically proved using a sequence of commands called tactics[2]. When Lean processes a proof code, it initially parses it into a syntax tree, then elaborates the tree into an expression, and finally sends the expression to the kernel to verify that it has the correct type, ensuring that the provided proof actually proves the claimed theorem. Specifically, the elaborator processes the sequence of tactics and constructs a proof term.

Extracting data from Lean source code is technically complex. Among recent projects that facilitate data extraction, notable examples include *lean-training-data* (Morrison, 2023a) and *LeanDojo* (Yang et al., 2023). Both projects are implemented in Lean and utilize Lean's internal structure known as InfoTree. The Lean elaborator uses this structure to store various pieces of information, such as intermediate proof states and positional information, which are later used in the user interface. This method offers the advantage of not requiring the recompilation of the source code; however, it also has two disadvantages. Firstly, certain internal temporal information that could be useful for training is not available. Secondly, this setup does not allow us to modify the proof. For a more reliable and customizable option, we

---

[1]Moscow Institute for Physics and Technology [2]Yandex [3]London Institute for Mathematical Sciences. Correspondence to: Vasilii Nesterov <vas.nesterov63@google.com>.

*AI for MATH Workshop at ICML 2024*. Copyright 2024 by the author(s)

---

[1]In our paper we work only with Lean version 4.6.0.

[2]For a list of tactics with descriptions, please refer to https://github.com/haruhisa-enomoto/mathlib4-all-tactics

chose to integrate tracing code into the Lean elaborator's source code. This approach enables us to apply various proof modifications and trace training data during the elaboration process, with full access to the elaborator's state. We release our extraction tool called **Lean4trace**.

Most prior studies (Polu et al., 2023; Lample et al., 2022; Yang et al., 2023) in theorem proving in Lean exclusively relied on human-written proofs from Mathlib as the main source of training data. Although it is the largest and highest-quality source of formalized math in Lean, the proofs in Mathlib are often compressed, meaning that we can potentially extract more than one proof state from a single tactic. For example, the most frequently used tactics in the library are `rw` and `simp`, both of which take a list of lemmas and apply them one by one. Thus, they can be replaced by a sequence of individual tactic applications. Even humans, when searching for a proof, would likely try to apply lemmas one by one. Therefore, such proof rewriting not only provides more training data for the model but also makes the data more meaningful and easier for the model to understand. This could be especially useful for retrieval models, as the model is then trained to retrieve lemmas that are conceptually related to the current state, rather than some intermediate one. We refer to this as **tactic decomposition data augmentation**.

Some tactics in Lean, such as `rw`, `apply`, and `exact`, perform basic operations corresponding to single reasoning steps. Others, which we refer to as 'automatic' in this paper, are capable of more complex reasoning. Examples of such tactics include `solve_by_elim`, which recursively discharges the goal using the local context; `tauto`, which derives the goal from hypotheses in propositional logic; and `aesop`, which performs tree-based proof search using a set of predefined rules. There are many other automatic tactics, including domain-specific ones. Although they are relatively rare in Mathlib, we found that these tactics can prove a notable fraction of goals. Several factors contribute to their limited presence in human-written proofs: some of them are relatively new, proofs using automatic tactics are less robust, and these tactics generate longer proof terms. Nonetheless, they appear useful for automated proof search. Therefore, we tested each automatic tactic against each proof state in the data and collected all successful examples. We refer to this process as **automatic tactic data augmentation**.

In summary, this paper makes the following contributions:

- We introduce **Lean4trace**, a novel tool for data extraction from Lean 4 source code, seamlessly integrated into the Lean elaborator. It enables interaction with existing proofs (e.g., testing automatic tactics) and extracts more proof states than previous extraction tools.

- We propose two methods of data augmentation in au-

tomated theorem proving: **automatic tactic data augmentation** and **tactic decomposition data augmentation**.

- We demonstrate that the proposed data augmentations enhance the performance of the ReProver model on Mathlib (+9.4% Pass@1) and MiniF2F (+9.1% Pass@1) benchmarks.

## 2. Related work

The available data for Lean is rather scarce but complex to learn as it requires advanced reasoning abilities. To tackle this problem, some papers try to improve, modify or enlarge training data. For example, in GPT-f (Polu et al., 2023) they used Expert Iteration to generate new proofs for the theorems from the training set using the model at the current iteration. Some of the generated proofs are shorter than the original ones. This makes the proof search faster as the model tends to find shorter proofs. In (Wu et al., 2022) they tried to mine new theorems with proofs by utilizing large language model (LLM). The idea is to take theorems in natural language and prompt LLM to translate them to formal language, and then search for the proofs using pre-trained theorem prover. As a result, they obtained larger dataset with theorems from different domain. Some papers generate synthetic data (most commonly, equalities and inequalities as it is easy in this case to generate the proof), e.g. (Polu et al., 2023), (Lample et al., 2022).

The GPT-f model was trained on additional proof artifacts collected as described in (Han et al., 2022). The artifacts are some artificially generated problems, for example, predicting missing proof term or type. Such data is not directly connected to proof generation but allows to pretrain the language model on formal language domain.

Another interesting idea was proposed in (Jiang et al., 2022). The authors modified the proofs by using hammers (Paulsson & Blanchette, 2012) where possible. As a result, the model trained on such data learns to call hammer and finds more proofs.

Most recent papers almost fully rely on LLM's capabilities to few-shot learning. In (Thakur et al., 2023) the pre-trained language model serves as an intellectual proof step generator. It is prompted to generate proof steps for the current proof state given previous attempts, error messages and retrieved lemmas.

While most of the previously described models generate proof step-by-step, there are few papers that attempt to generate the whole proof at once. In (Jiang et al., 2023) they assumed that each formal theorem is equipped with informal statement. Having this they first prompt LLM to generate informal proof sketch, i.e. high-level proof

plan in natural language, and then use SledgeHammer to proof individual high-level steps in the proof. The papers (Zheng et al., 2023; First et al., 2023) prompts LLM to interact with the proof assistant in a chat manner. The idea is to provide feedback to the language model to fix the error in the proof. The work (Xin et al., 2023) follows (Jiang et al., 2023) assuming that each formal statement has its corresponding informal statement and informal proof. They decompose the proof into steps, formalize and prove each step using LLM and evolving library of skills (verified lemmas database, problems statements and newly generated lemmas by prover).

The authors of (Azerbayev et al., 2023) collect large dataset of math and code related texts and train LLM called Llemma. The model can be used to solve various math problems beyond formal theorem proving.

The vast majority of recent papers rely on large language models. While they are very useful at generating proof plans, they can be too expensive to solve individual steps. This can be done more efficiently with smaller language models, like in (Yang et al., 2023), as we can generate and check more tactics at the same amount of time. To improve such models, we aim to generate larger and simpler training dataset.

## 3. Methods

### 3.1. Experimental setup and baseline

The main goal of our paper is to propose and evaluate various training data modifications for automated theorem proving in Lean. We chose LeanDojo and ReProver, proposed in (Yang et al., 2023), as the baseline data extraction tool and prover, respectively, for two reasons: the experiments do not require large computational resources, and the code is open-sourced, making it easy to build upon. Note, however, that we use LeanDojo only for interaction with Lean, while utilizing **Lean4trace** for data extraction.

In our experiments we follow the simplified pipeline of ReProver (Yang et al., 2023). We fine-tune the ByT5-small model (Xue et al., 2022) on the data extracted from Mathlib. The model is trained to generate a proof step, i.e., a single tactic application, conditioned on a proof state that includes the local context—information that appears in the InfoView as a user proves the theorem. Note, that in ReProver the prover consists of 2 models: tactic generator and retriever that tries to retrieve relevant lemmas. However, in our work use only tactic generator. We evaluate the model on the LeanDojo benchmark, which consists of 2,000 randomly selected theorems from Mathlib, and on the MiniF2F benchmark (Zheng et al., 2021). During the proving process, the model generates multiple tactic candidates at each step, which are used in a standard best-first search algorithm to find a proof with a cumulative log-prob as a ranking criterion, see details in (Polu & Sutskever, 2020). We evaluate the Pass@1 metric: the fraction of theorems which can be proven by the prover within 10 minutes in one attempt.

### 3.2. Data extraction in Lean

The elaborator is a part of the Lean system that infuses syntactic objects with meaning. In particular, the elaborator processes tactic blocks, applying tactics and constructing a proof term as a result. We modify the Lean elaborator so that at each tactic invocation, it traces:

- The current proof state, obtained from the elaborator's state.

- The proof step (tactic) that the elaborator is going to process.

- The used premises, which we extract by finding all named constants in the proof step.

Additionally, we trace some meta information such as the name of the file, module, theorem being processed, and the position of the proof step.

To enable the use of **Lean4trace** for generating training data for retrieval-augmented models as described in (Yang et al., 2023), we take the following steps. First, we utilize the **import-graph** (Morrison, 2023b) tool to extract the import structure, which is necessary for determining which lemmas are accessible within a given theorem. Finally, to build the corpus of all declared definitions that can be used as premises, we use the **lean-training-data** (Morrison, 2023a) tool.

From the traced data, we build a *canonical* dataset, where "canonical" means that this data is obtained from the original human-written proofs for further comparison with augmented data. It contains all proof states visible to the user.

### 3.3. Tactic decomposition data augmentation

To test if tactic decomposition can help the model learn better, we focus on the two most frequent tactics in Mathlib: `simp` and `rw`. Both of these tactics take a list of rules as an argument but process them differently. The `rw` tactic takes a list of rules and rewrites the goal by applying them in a given order. Meanwhile, the `simp` tactic, given a set of rules, applies them in an order determined by its heuristics and can apply each rule multiple times. Therefore, we decompose proof steps containing these tactics in different ways, which we describe below. These two tactics are the most frequent tactics in Mathlib, and in most cases they are applied to the list of multiple rules, so their decomposition gives a notable amount of new data.

The decomposition procedure for `rw` is straightforward: we replace a proof step of the form `rw [h₁,...,hₙ]` with a sequence of proof steps `rw [h₁],...,rw [hₙ]`. The `rw` tactic is implemented as a thin wrapper over `rewrite`, which attempts to close the goal with `rfl` at the end. To decompose `rw`, we modify the code of `rewrite` so that when it takes a list of rules containing more than one rule, it applies single-rule `rw` multiple times, tracing intermediate proof states. As a result, our modification affects `rw` and some of its variations, such as `erw` and `nth_rw`.

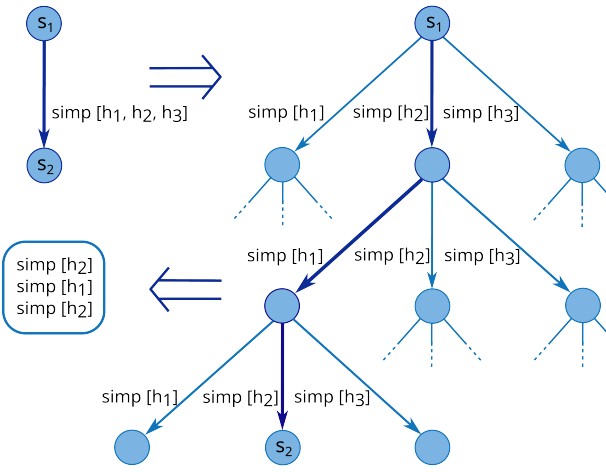

*Figure 1.* Process of expanding the `simp` tactic. In the original proof, the proof state $s_2$ is obtained by applying `simp [h₁, h₂, h₃]` in state $s_1$. Using BFS, we find a sequence `simp [h₂]; simp [h₁]; simp [h₂]` which also leads to $s_2$. In this example, $h_2$ is used twice, and $h_3$ can be omitted. Such situations actually occur in Mathlib proofs.

The case of `simp` is more complex because the order of rule applications is not determined by the user. While the order could be extracted after the tactic's invocation using Lean meta-programming, we use a different approach here. Given the list of rules, we employ a Breadth-First Search algorithm, treating proof states as vertices with edges corresponding to single-rule `simp` applications with rules from the list, see the Figure 1. Once a sequence leading to the target proof state is found, the search stops, and the proof states forming the sequence are traced. This method has the additional advantage of finding the shortest such sequence.

### 3.4. Automatic tactics data augmentation

To train our model to use automatic tactics, we mine additional data using the following procedure: at each tactic invocation, before applying the original tactic, we attempt to apply tactics from a fixed list of automatic tactics. If the application is successful, meaning the tactic has closed the goal, we trace this automatic proof step. If multiple

automatic tactics are able to close the goal, we trace all of them.

In some cases, a single automatic tactic can replace a sequence of multiple tactics in the original proof. Typically, the automatic tactic can close the goal from each intermediate proof state. Instead of keeping only the first automatic tactic application that closes the goal, we retain all these data points to ensure that the model learns when the automatic tactic is applicable.

Sometimes, tactics can take a very long time or even hang indefinitely. To address this, we set a time limit of 10 seconds for each automatic tactic application. Since Lean 4 does not yet support timeouts internally, we use an external Python script as a temporary solution. This script monitors the building process and restarts it if it gets stuck in a proof state, adding the problematic proof state to a blacklist. Proof states on the blacklist are not tested again.

## 4. Experiments and results

### 4.1. Dataset's statistics

Firstly, to give an overview of our dataset, we provide the frequencies of the most popular tactics in Mathlib in Table 1. Notably, `rw` and `simp` cover more than 28% of all proof states extracted from Mathlib, which motivates us to focus on them in tactic decomposition augmentation.

| Tactic | Frequency, % |
|---|---|
| rw | 15.2 |
| simp | 13.1 |
| · (cdot) | 10.1 |
| exact | 9.8 |
| have | 5.1 |
| apply | 3.6 |
| refine´ | 3.4 |
| intro | 3.0 |
| simpa | 2.1 |
| rfl | 2.0 |
| ext | 1.9 |
| obtain | 1.8 |
| rintro | 1.8 |
| rcases | 1.6 |
| dsimp | 1.4 |
| simp_rw | 1.4 |
| cases | 1.3 |
| refine | 1.1 |
| let | 1.0 |

*Table 1.* Most frequent tactics in Mathlib

Next, we present statistics on the usability of each automatic tactic in Table 2. Note that the `aesop` tactic alone can close more than 20% of proof states in Mathlib. Unfortunately, different general-purpose automatic tactics tend to close

similar sets of goals. In total, the automatic tactics we used can close 23.6% of proof states. This demonstrates that such tactics are quite powerful, and any prover could benefit from using them. Additionally, they can be considered as a baseline in automated theorem proving in Lean.

| Automatic tactic | Solved goals, % |
|---|---|
| aesop | 21.8 |
| simp_all | 16.6 |
| simp [*] | 13.5 |
| simp_arith | 9.6 |
| tauto | 8.9 |
| solve_by_elim | 7.8 |
| continuity | 5.7 |
| norm_num | 5.6 |
| abel | 1.5 |
| omega | 1.4 |
| nlinarith | 1.1 |
| linarith | 1.0 |
| ring | 0.8 |
| decide | 0.6 |
| group | 0.5 |

*Table 2.* Number of goals can be solved by auto tactics.

## 4.2. Comparison with LeanDojo extraction

We compare our approach with LeanDojo, our main baseline. The number of extracted proof states and the resources required for each augmentation are provided in Table 3. We refer to the dataset extracted by our tracing without augmentations as **canonical**. Note that our approach requires considerably less RAM, making it possible to run on a regular PC. The key factor that slows down our data augmentation mining is the lack of timeout mechanisms in Lean 4, the implementation of which would greatly speed up the process.

The difference in the number of traced proof states between the LeanDojo approach and ours mainly comes from tactic combinators. For example, the construction $tac_1$ <;> $tac_2$ in Lean means that $tac_2$ should be applied to all goals produced by $tac_1$. In LeanDojo tracing, such a construction is treated as a single proof step, while we trace separate proof steps for each produced goal.

The last three rows in Table 3 refer to augmentations, which we mix with the canonical subset during training. The provided number of proof states represents the number of **new** proof states in relation to the canonical subset.

---

[3]The time is measured using 32-core CPU.

[4]This does not include time spent for building Mathlib.

[5]LeanDojo tracing requires 48 GB even when running on a single core.

| Dataset | # proof states | Time[3] | RAM, GB |
|---|---|---|---|
| LeanDojo tracing | 273k | 1 h [4] | 48[5] |
| Canonical | 352k | 31 min | 17 |
| rw decomposition | 110k | 34 min | 18 |
| simp decomposition | 37.7k | 11 h | 24 |
| Automatic tactics | 318k | 7 days | 10 |

*Table 3.* Resources required for tracing.

## 4.3. Theorem proving

In this subsection we discuss models trained on augmented datasets and their proving capabilities. The Pass@1 metric is provided in Table 4 (note, that the numbers are for the model without retrieval). Automatic tactics augmentation provides +1.7% improvement over canonical tracing, while tactics decomposition gives +2% improvement on Mathlib dataset.

| Model & training data | Mathlib | MiniF2F |
|---|---|---|
| ReProver | | |
|   LeanDojo data[6] | 48.6 | 26.5[7] |
|   Canonical | 56.3 | **35.6** |
|   Canonical + Tactics decomposition | **58.0** | 30.0 |
|   Canonical + Automatic tactics | 57.6 | 33.6 |
| Thor + expert iteration (Wu et al., 2022) | | 35.2 |
| COPRA + GPT-4 (Thakur et al., 2024) | | 30.7 |
| Thor (Jiang et al., 2022) | | 29.9 |
| Lean Expert Iteration (Polu et al., 2023) | | 29.6 |

*Table 4.* Pass@1 for theorem proving.

In addition to Mathlib, we test our models on a test subset of the MiniF2F benchmark, which consists of formalized Olympiad-level mathematics problems. A notable feature of this benchmark is that its Lean 4 version was released after all the data the model was trained on (including pre-training) had been gathered. Therefore, there are no Lean 4 proofs for MiniF2F available on the internet so far, ensuring that the model has never encountered these proofs during training.

There is a notable increase in metrics on MiniF2F, even though no extra fine-tuning was performed specifically for it. We achieve our best results with the model trained on data without augmentations: it proves 87 of the 244 theorems presented in the MiniF2F test subset. We examined the successful proofs found by the model and discovered that most of them rely on automatic tactics (see the Figure 2).

---

[6]The results are taken from the paper (Yang et al., 2023) and pertain to a model that also relies on an additional model, which retrieves potentially useful auxiliary theorems and definitions at each step.

[7]This results were obtained using the Lean 3 version of the MiniF2F benchmark.

This demonstrates that modern tools for proof automation in Lean are powerful enough to enable a model equipped with them to prove a significant portion of the dataset with minimal inherent reasoning.

At the same time, the quality degrades when we apply augmentations. This requires further investigation, but it might be explained as follows. The domain of MiniF2F problems differs from that of Mathlib, and when augmentations are applied, the proportion of proof steps that are useful for MiniF2F decreases: for example, `simp` is very rare in found proofs for MiniF2F but occur frequently in both augmentations. In contrast, `norm_num` is widely used tactic in MiniF2F which is rare in Mathlib. This reduction in relevant proof states could lead to the observed degradation in performance.

For comparison we also provide the Pass@1 metric on MiniF2F reported in prior studies.

## 5. Conclusion

In this paper, we present **Lean4trace**, a novel tool for data extraction and augmentation tailored for training neural theorem provers in Lean. Our experimental results demonstrate that models trained using our dataset achieve a 9% higher performance on the MiniF2F benchmark compared to Re-Prover (Yang et al., 2023), when trained and evaluated under identical conditions. While proposed augmentations provides improvements on Mathlib dataset, they may degrade the model when evaluated on a dataset from different distribution. Nevertheless, our tool allows gathering a more complete set of proof states in canonical setup and significantly reduces computational resource requirements compared to LeanDojo (Yang et al., 2023), making it feasible to run on a modern PC. We believe that these advancements will lower the barrier to entry in this field, fostering more accessible and widespread research in neural theorem proving.

## 6. Acknowledgements

This work was supported by a grant for research centers in the field of artificial intelligence, provided by the Analytical Center for the Government of the Russian Federation in accordance with the subsidy agreement (agreement identifier 000000D730324P540002) and the agreement with the Moscow Institute of Physics and Technology dated November 1, 2021 No. 70-2021-00138.

## References

Azerbayev, Z., Schoelkopf, H., Paster, K., Dos Santos, M., McAleer, S., Jiang, A. Q., Deng, J., Biderman, S., and Welleck, S. Llemma: An open language model for mathematics. *arXiv preprint arXiv:2310.06786*, 2023.

```
theorem mathd_numbertheory_135
    (n A B C : Nat)
    (h₀ : n = 3^17 + 3^10)
    (h₁ : 11 | (n + 1))
    (h₂ : [A,B,C].Pairwise (· ≠ ·))
    (h₃ : {A,B,C} ⊆ Finset.Icc 0 9)
    (h₄ : Odd A ∧ Odd C)
    (h₅ : ¬ 3 | B)
    (h₆ : Nat.digits 10 n = [B,A,B,C,C,A,C,B,A]) :
    100 * A + 10 * B + C = 129 := by
  aesop -- general-purpose automatic tactic

theorem mathd_numbertheory_229 :
    (5^30) % 7 = 1 := by
  decide -- tactic that proves some "decidable" goals
      -- here it just computes (5^30) % 7

theorem mathd_algebra_33
    (x y z : Real)
    (h₀ : x ≠ 0)
    (h₁ : 2 * x = 5 * y)
    (h₂ : 7 * y = 10 * z) :
    z / x = 7 / 25 := by
  field_simp [h₀, h₁]
  linarith -- tactic that solves linear equations
```

*Figure 2.* Examples of proofs found by our model. Presented proofs almost completely rely on automatic tactics.

Barras, B., Boutin, S., Cornes, C., Courant, J., Filliâtre, J.-C., Giménez, E., Herbelin, H., Huet, G., Muñoz, C., Murthy, C., Parent-vigouroux, C., Paulin-Mohring, C., Saïbi, A., and Werner, B. The Coq proof assistant reference manual : Version 6.1. 06 1997.

de Moura, L. M., Kong, S., Avigad, J., van Doorn, F., and von Raumer, J. The lean theorem prover (system description). In *CADE*, 2015. URL https://api.semanticscholar.org/CorpusID:232990.

First, E., Rabe, M., Ringer, T., and Brun, Y. Baldur: Whole-proof generation and repair with large language models. In *Proceedings of the 31st ACM Joint European Software Engineering Conference and Symposium on the Foundations of Software Engineering*, pp. 1229–1241, 2023.

Han, J. M., Rute, J., Wu, Y., Ayers, E. W., and Polu, S. Proof artifact co-training for theorem proving with language models. In *International Conference on Learning Representations*, 2022.

Jiang, A. Q., Li, W., Tworkowski, S., Czechowski, K., Odrzygóźdź, T., Miłoś, P., Wu, Y., and Jamnik, M. Thor: Wielding hammers to integrate language models and automated theorem provers. *Advances in Neural Information Processing Systems*, 35:8360–8373, 2022.

Jiang, A. Q., Welleck, S., Zhou, J. P., Li, W., Liu, J., Jamnik, M., Lacroix, T., Wu, Y., and Lample, G. Draft, Sketch,

and Prove: Guiding formal theorem provers with informal proofs. In *International Conference on Learning Representations*, 2023. URL https://doi.org/10.48550/arXiv.2210.12283.

Lample, G., Lacroix, T., Lachaux, M.-A., Rodriguez, A., Hayat, A., Lavril, T., Ebner, G., and Martinet, X. Hypertree proof search for neural theorem proving. *Advances in neural information processing systems*, 35:26337–26349, 2022.

mathlib Community, T. The lean mathematical library. In *Proceedings of the 9th ACM SIGPLAN International Conference on Certified Programs and Proofs*. ACM, jan 2020. doi: 10.1145/3372885.3373824. URL https://doi.org/10.1145%2F3372885.3373824.

Megill, N. and Wheeler, D. A. *Metamath: a computer language for mathematical proofs*. Lulu. com, 2019.

Morrison, K. lean-training-data. https://github.com/semorrison/lean-training-data, 2023a.

Morrison, K. import-graph. https://github.com/leanprover-community/import-graph, 2023b.

Nipkow, T., Paulson, L., and Wenzel, M. *Isabelle/HOL — A Proof Assistant for Higher-Order Logic*. 01 2002. doi: 10.1007/3-540-45949-9.

Paulsson, L. C. and Blanchette, J. C. Three years of experience with sledgehammer, a practical link between automatic and interactive theorem provers. In *Proceedings of the 8th International Workshop on the Implementation of Logics (IWIL-2010), Yogyakarta, Indonesia. EPiC*, volume 2, 2012.

Polu, S. and Sutskever, I. Generative language modeling for automated theorem proving. *arXiv preprint arXiv:2009.03393*, 2020.

Polu, S., Han, J. M., Zheng, K., Baksys, M., Babuschkin, I., and Sutskever, I. Formal mathematics statement curriculum learning. In *The Eleventh International Conference on Learning Representations*, 2023. URL https://openreview.net/forum?id=-P7G-8dmSh4.

Thakur, A., Wen, Y., and Chaudhuri, S. A language-agent approach to formal theorem-proving. *arXiv preprint arXiv:2310.04353*, 2023.

Thakur, A., Wen, Y., and Chaudhuri, S. A language-agent approach to formal theorem-proving, 2024. URL https://openreview.net/forum?id=XCMbagV0No.

Vaswani, A., Shazeer, N., Parmar, N., Uszkoreit, J., Jones, L., Gomez, A. N., Kaiser, L., and Polosukhin, I. Attention is all you need, 2023.

Wu, Y., Jiang, A. Q., Li, W., Rabe, M., Staats, C., Jamnik, M., and Szegedy, C. Autoformalization with large language models. *Advances in Neural Information Processing Systems*, 35:32353–32368, 2022.

Xin, H., Wang, H., Zheng, C., Li, L., Liu, Z., Cao, Q., Huang, Y., Xiong, J., Shi, H., Xie, E., et al. Lego-prover: Neural theorem proving with growing libraries. *arXiv preprint arXiv:2310.00656*, 2023.

Xue, L., Barua, A., Constant, N., Al-Rfou, R., Narang, S., Kale, M., Roberts, A., and Raffel, C. ByT5: Towards a token-free future with pre-trained byte-to-byte models. *Transactions of the Association for Computational Linguistics*, 10:291–306, 2022. doi: 10.1162/tacl_a_00461. URL https://aclanthology.org/2022.tacl-1.17.

Yang, K., Swope, A. M., Gu, A., Chalamala, R., Song, P., Yu, S., Godil, S., Prenger, R., and Anandkumar, A. LeanDojo: Theorem proving with retrieval-augmented language models, 2023.

Zheng, C., Wang, H., Xie, E., Liu, Z., Sun, J., Xin, H., Shen, J., Li, Z., and Li, Y. Lyra: Orchestrating dual correction in automated theorem proving. *arXiv preprint arXiv:2309.15806*, 2023.

Zheng, K., Han, J. M., and Polu, S. minif2f: a cross-system benchmark for formal olympiad-level mathematics. In *International Conference on Learning Representations*, 2021.

