# OpenReview forum: "Lean4trace: Data augmentation for neural theorem proving in Lean"
_ICML.cc/2024/Workshop/AI4MATH — ICML 2024 Workshop AI4MATH Poster_

### Official Review · Reviewer_2txq · 2024-06-12

**Rating:** 6
**Confidence:** 5

**Summary:**

This paper proposes a method of data augmentation for theorem proving data in the formal language Lean 4. Two methods are used: 1. Tactic decomposition: decompose tactics such as simp and rw to get intermediate tactic states and proofs steps; 2. Automatic tactics: apply automatic tactics such as aesop at each proof step. The authors also improve LeanDojo's tracing method to curate a dataset called canonical. The ByT5 model trained using these data achieves +9.4% improvement on pass@1 on Mathlib and +9.1% on miniF2F.

**Questions:**

1. Is the data tracing technique used in canonical based on LeanDojo code? The part on canonical part is not well explained in the paper. Please improve the explanations.
2. In line 193, you mentioned "Note that our approach requires considerably less RAM". Why is this? And can you give a concrete quantity for the RAM consumption of both methods?
3. Are you going to open-source your code and dataset?

**Reasons To Accept:**

1. The proposed method is well motivated since data augmentation is a very important direction for formal method due to data scarcity;
2. The data augmentation techniques are well explained in the paper.
3. The baselines and the approach are evaluated on standard benchmarks in a fair way.
4. The performance improvements are considerable on Mathlib and miniF2F.

**Reasons To Reject:**

1. The part on canonical is not well explained, while the performance improvements are mainly due to canonical dataset;
2. The contribution on "data augmentation" as claimed in the title of paper seems to harm the performance on many scenarios.
3. The novelty is limited as the proposed techniques focus on rather microscopic aspects and could be specific to Lean 4 language and it's hard to generalize.

---

### Official Review · Reviewer_xzY5 · 2024-06-12

**Rating:** 7
**Confidence:** 4

**Summary:**

This paper proposes a novel data augmentation method for tracing the Lean Mathlib library. The method consists of two main components:
- Decomposing composite proof steps into multiple simpler steps, including the decomposition of rw and simp tactics.
- Using automatic tactics to prove theorems and obtain new proof state/tactic pairs.

The experimental results show that the newly traced data provides a significant improvement on the Lean Mathlib dataset. Additionally, the new tracing method achieves reduced memory consumption.

**Questions:**

Please justify the question I raised in the `Reasons To Reject`.

**Reasons To Accept:**

- The proposed augmentation is novel and interesting, offering a new tracing technique that extracts significantly more proof states than LeanDojo, with less memory consumption and faster extraction time.
- The experimental results are promising, particularly with the canonical data.
- The paper is well-written and easy to follow.

**Reasons To Reject:**

- The performance gain of the newly proposed augmentation methods is marginal. In the MathLib dataset, the automatic method even has negative effects.
- The majority of the performance gain is achieved with the canonical dataset, which is not discussed in detail in the paper. It would be helpful to provide a more detailed explanation of this canonical dataset.

---

### Official Review · Reviewer_SCYZ · 2024-06-12

**Rating:** 4
**Confidence:** 3

**Summary:**

The paper introduces Lean4trace, an open-source tool for extracting training data from Lean 4 sources, addressing the challenge of limited training data in integrating large language models with theorem provers. By deeply integrating with the Lean elaborator, Lean4trace enables real-time proof modifications and traces proof states, steps, and premises, along with additional meta information. The authors propose two data augmentation methods: decomposing complex proof steps of frequent tactics like simp and rw into simpler steps, and testing automatic tactics at each proof state to collect successful applications.These methods significantly enhance training data, resulting in models that successfully prove 58.0% of theorems from a hold-out Mathlib subset and 35.6% of the MiniF2F benchmark test subset, demonstrating the effectiveness of the approach.

**Questions:**

- Could you provide more detailed descriptions of the process for extracting proof information using the Lean elaborator? Consider adding details with step-by-step examples demonstrating the extraction of proof information and subsequent modifications.
- Could you offer a more detailed analysis of how the augmentations impact the model's performance on different types of proofs? For instance, how do they affect the pattern of tactic application and proof structure?
- What are the possible reasons for the decline in performance on the MiniF2F benchmark? How do you plan to address the issue of the model becoming more attuned to specific scenarios other than Mathlib?

**Reasons To Accept:**

**Originality**: The paper shows originality by modifying the Lean elaborator to extract detailed information about tactic invocations, enabling real-time modifications and comprehensive data extraction. The introduction of Lean4trace as an open-source tool and the combination of Tactic Decomposition Data Augmentation and Automatic Tactics Data Augmentation highlight the innovative approach.

**Quality**: The work's quality is evident through its rigorous methodology and significant performance improvements: from 48.6% to 56.3% on Mathlib and from 26.5% to 35.6% on MiniF2F. Detailed tracing of proof states and the use of external tools for managing timeouts reflect a useful approach. Data augmentation techniques further boosted Mathlib performance to 58.0%/57.6%.

**Clarity**: The paper is well-written and structured, making it accessible.

**Significance**: This paper proposes a data augmentation method to solve the data scarcity problem when using language models to perform theorem proving problems. It deeply expands the existing methods, better interacts with the internal state of the proof assistant's verifier, and provides new tool options for the community. The article explores some simple proof step replacement methods to explore equivalent but easier to understand proof steps, providing new ideas for data augmentation for theorem proving.

**Reasons To Reject:**

**Lack of Detail and Examples**: The section on extracting proof information is inadequately detailed, and the absence of concrete examples makes it challenging to grasp the practical implications and effectiveness of the proposed method.

**Insufficient Analysis of Proof Behavior**: There is a notable lack of in-depth analysis regarding the model's proof behavior post-data augmentation. The paper does not sufficiently explore how the augmentations impact the model's performance, leaving readers with an incomplete understanding of their efficacy.

**Engineering Tricks**: From the current description of the article, the expansion of the premise used by the tactic and the replacement of the advanced automated tactic are more of an engineering trick, and at most provide a rewrite and trivial replacement of the existing proof to make it more understandable, and it is difficult to provide different proofs with essential differences.

**Limited Improvement in Proof Ability**: The augmentations do not significantly enhance the model's proof capabilities. This is evident from the pass rate control experiments, which show no substantial improvement. In fact, performance on the MiniF2F benchmark declined, suggesting that the model may have become more attuned to specific scenarios in Mathlib without achieving broader proof proficiency.

---

### Official Review · Reviewer_uB1G · 2024-06-12

**Rating:** 6
**Confidence:** 5

**Summary:**

This paper introduces **Lean4trace**, a novel tool designed for data extraction and augmentation specifically for training neural theorem provers within the Lean theorem proving environment. **Lean4trace** is deeply integrated with the Lean elaborator, enhancing its functionality by enabling on-the-fly interaction with existing proofs and extracting more comprehensive proof states than prior tools.

**Questions:**

1. I am skeptical about replacing automatic tactics in any row of tactics, because there are some high-level tactics, such as tidy, which can directly replace several consecutive rows of tactics.
2. When searching for the sub-level tactic sequence corresponding to Simp, is the language model used for Breadth-First Search?

**Reasons To Accept:**

1.This approach utilizes the Lean4 interpreter to develop two simple methods for extracting more proof states and tactics.
2.Developed an automatic detection tool to solve the problem of timeout in Automatic tactics during proof

**Reasons To Reject:**

1.This method is not novel in my opinion. It just applies some tactics' sub-level proofs to the theorem prover to get more detailed proof state and tactic.
2.This is more like a method[1][2] of breaking down the proof goal and then proving the sub-goals, but this idea is not novel.

**Reference**
[1] Lample, Guillaume, et al. "Hypertree proof search for neural theorem proving." Advances in neural information processing systems 35 (2022): 26337-26349.
[2] Zhao X, Li W, Kong L. Decomposing the enigma: Subgoal-based demonstration learning for formal theorem proving[J]. arXiv preprint arXiv:2305.16366, 2023.

---

### Meta-Review · Area_Chair_9brH · 2024-06-13

**Recommendation:** Accept (Poster)
**Confidence:** 4

**Metareview:**

This paper answers a question that is often asked in Lean theorem proving: how do models improve when having data when tactics like simp are split in detailed steps. As the reviewers suggest there are several good points and some limitations, in particular there are important details missing (extracting proof information, canonical datasets, etc.). Surprisingly, the automatic method can sometimes have negative effects. While it would be important to understand why, this is not an obstacle for publication in my opinion, given that the idea explored in this paper is natural and was worth to be explored anyway. I recommend accepting, taking a leap of faith, that the authors will at least elaborate on the missing details pointed by the reviewers in the camera-ready version.

---

### Decision · Program_Chairs · 2024-06-13

Accept (Poster)